# IN-CONTEXT LEARNING FOR FEW-SHOT MOLECULAR PROPERTY PREDICTION

## ABSTRACT

In-context learning has become an important approach for few-shot learning in Large Language Models because of its ability to rapidly adapt to new tasks without fine-tuning model parameters. However, it is restricted to applications in natural language and inapplicable to other domains. In this paper, we adapt the concepts underpinning in-context learning to develop a new algorithm for few-shot molecular property prediction. Our approach learns to predict molecular properties from a context of (molecule, property measurement) pairs and rapidly adapts to new properties without fine-tuning. On the FS-Mol and BACE molecular property prediction benchmarks, we find this method surpasses the performance of recent meta-learning algorithms at small support sizes and is competitive with the best methods at large support sizes.

## 1 INTRODUCTION

In-context learning describes an emergent property of large language models (LLMs) that enables them to solve new tasks from only a few demonstrations and without any gradient updates to the model parameters (Brown et al., 2020). This capacity to rapidly adapt to new tasks contrasts sharply with typical few-shot learning algorithms that either use gradient updates, or distance computations to prototypical class centroids, to adapt the pre-trained model to the few-shot learning objective. As a result, in-context learning has become a powerful approach for few-shot learning applications in natural language; however, it is inapplicable to other domains as it uses a language modeling objective to train the model.

One such domain is molecular science where few-shot learning is critical to drug discovery. After a biological target has been identified, finding small molecules that inhibit this target may lead to desirable outcomes. For example, inhibiting the protein 15-PGDH with a small molecule inhibitor leads to rejuvenation of aged skeletal muscle tissue in animal studies, effectively reverse-aging the cells (Palla et al., 2021). However, identifying small molecules that inhibit a specific biological target or pathway can be difficult as the number of experimental measurements collected for a given target or pathway is often on the order of 10s or 100s of measurements. Moreover, collecting additional experimental measurements is costly and time-intensive, often entailing a post-doc in a laboratory searching the literature for inhibitors to similar targets, ordering reagents, and running biological assays with limited throughput (10s of molecules per assay). Accordingly, there is great interest in applying few-shot learning methods to molecular property prediction, and in particular high-throughput screening where small molecule libraries of order $10^9$ can be processed to identify promising inhibitors.

In this work, we develop an algorithm that brings the benefits of in-context learning — namely predicting the label of a point from a context of demonstrations in a single forward pass — to molecular property prediction. Our approach, called Context Aware Molecule Prediction (CAMP), learns to predict molecular properties from a context of (molecule, property) demonstrations and can rapidly adapt to new tasks without finetuning.

CAMP operates by first encoding query and demonstration molecules with a molecule encoder and property measurements with a label encoder. Molecule and label embeddings are concatenated together, and we feed the resulting sequence of vectors into a Transformer encoder that learns to predict the label of the query from the context of demonstrations. This approach is visualized in Figure 1. Crucially, and similar to in-context learning, CAMP does not use a fine-tuning step to

update the models parameters — or a post-processing step to compute prototypical class centroids — but rather predicts a molecule's property directly from the input demonstrations and query molecule.

In summary, our primary contribution is to develop an in-context learning algorithm for molecular property prediction. This is not a trivial adaptation. It requires recasting meta-learning as a sequence modeling paradigm that trains from random initialization on molecular property prediction datasets and is invariant to the order of examples within this sequence. On two molecular property prediction benchmarks, our empirical analysis finds this approach outperforms recent few-shot learning baselines at small support sizes and is competitive with the best baselines at large support sizes. Further, CAMP has low-latency, positioning it for applications in pharmacological discovery, such as high-throghuput screening where small molecule libraries of order $10^9$ are processed to identify promising inhibitors.

## 2 RELATED WORK

**Few-Shot Learning.** Few-shot learning refers to a class of techniques that aim to maximize model performance when supervised training data is limited. The few-shot training examples are referred to as the *support set* and few-shot dataset test examples are called the *query set*. A model is typically meta-trained on a large-scale dataset consisting of many different tasks, fine-tuned on the support set, and then evaluated on the query set. While there is significant diversity among few-shot methods, almost all can be categorized as *gradient-based* or *metric-based* algorithms (Bouniot et al., 2022).

Gradient-based algorithms (Finn et al., 2017; Wang et al., 2021; Raghu et al., 2019; Jamal & Qi, 2019; Finn et al., 2018) develop learned representations that can be quickly adapted to a new task by applying one or more gradient updates. Many approaches leverage a bi-level optimization loop in pre-training that maximizes model performance, not in the current step, but rather a future step by differentiating through each intermediary parameter update. This future step performance is then realized by finetuning the model's paramters on the support set. On the other hand, metric-based algorithms (Snell et al., 2017; Simon et al., 2020; Allen et al., 2019; Sung et al., 2018; Bateni et al., 2020) fix a distance metric (i.e. Euclidean distance, Mahalanobis distance, etc.) and learn a mapping from the input to an embedding space so that points belonging to the same class are close together and points belonging to different classes are far apart. Many approaches aggregate points belonging to the same class in the support set into a class-representative centroid, and classify points in the query set by choosing the label of the centroid with minimal distance to the query. Our approach fundamentally differs from both groups, and we formalize this difference as belonging to a new category of *context-based* algorithms.

**In-Context Learning.** In-Context learning refers to the process by which a LLM auto-regressively generates text when conditioned on a prompt of demonstrations and a query (Radford et al., 2019; Brown et al., 2020). Formally, given pairs of (example, solution) demonstrations as natural language $(x_1, y_1), (x_2, y_2), ..., (x_k, y_k)$ representative of the support set, and a query $x_{k+1}$, the solution to the query is generated auto-regressively as a next-word prediction $P(y_{k+1}|x_{k+1}, (x_k, y_k), ..., (x_1, y_1))$.

While LLMs are trained with a language modeling objective (Mnih & Hinton, 2008) or masked language modeling objective (Devlin et al., 2018), a plethora of recent work has focused on developing fine-tuning paradigms that use task-specific instructions or in-context prompts to improve few-shot performance (Wei et al., 2021; Min et al., 2021b; Chen et al., 2021). Other methods find few-shot performance to be highly sensitive to the order of demonstrations or phrasing of the prompt, and formulate protocols for *prompt-tuning*, or rephrasing the demonstrations into a near-perfect prompt, to reduce the variance of predictions for a given query (Arora et al., 2022; Schick & Schütze, 2020; Liu et al., 2021; Gao et al., 2020; Jiang et al., 2020). Nevertheless, even with these improvements, in-context learning performance typically falls behind that of gradient-based and metric-based few-shot learning algorithms (Schick & Schütze, 2020; Liu et al., 2022; Min et al., 2021b; Perez et al., 2021).

## 3 CONTEXT AWARE MOLECULE PREDICTION

In this work, we adapt the ideas underpinning in-context learning — namely learning to predict the label of a point from a context of demonstrations in a single forward pass — to molecular property prediction. However dissimilar from applications in natural language, an in-context learning algorithm for few-shot molecular property prediction should:

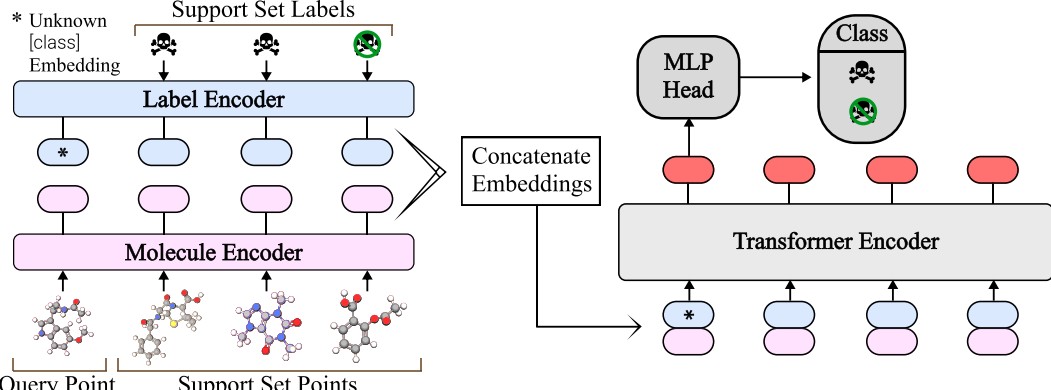

Figure 1: Overview of CAMP. Query and support set molecules are encoded with a MPNN and then concatenated with their corresponding label embeddings. We feed the resulting sequence of concatenated vectors into a Transformer encoder and extract the transformed query vector from the output sequence to predict its label.

§1 Handle molecular features and property measurement labels.
§2 Be invariant to the order of demonstrations in the context.
§3 Train from random initialization on molecular property prediction datasets.

To this end, we represent a demonstration as the concatenation of a learnable molecule embedding and a learnable label embedding. We use an MPNN (Gilmer et al., 2017) to encode molecules represented as a connected graph of atom-level features and a linear projection to encode one-hot class labels. Aggregating demonstrations across a new dimension, i.e. the sequence dimension, then forms a context.

The query molecule is similarly embedded and concatenated to a learnable unknown [class] label embedding to form a query vector. It is inserted into the sequence forming the context, and this sequence is then fed into a Transformer encoder to learn representations that are conditioned on both the context and query. Finally, the transformed query is extracted from the output of the Transformer encoder and subsequently passed through a shallow MLP to predict its label. A visual depiction of this approach for molecules with toxicity labels is shown in Figure 1. By construction, this formulation satisfies criteria §1, namely the capacity to handle molecular features and property measurement labels, and our findings in Section 6 empirically support §3. Similarly, in Section 4 we show §2 holds.

CAMP presents several advantages over typical in-context learning. Foremost among them, the molecules and property measurements do not need to be translated into natural language. Instead one can reuse molecule-specific encoders, such as MPNNs (Gilmer et al., 2017) or Graph Transformers (Mialon et al., 2021; Chen et al., 2022a; Dwivedi & Bresson, 2020; Maziarka et al., 2020; Honda et al., 2019), to learn molecule embeddings. Moreover, while typical in-context learning requires access to a 100 billion parameter pre-trained LLM, CAMP can be trained directly from random initialization on molecule property prediction datasets and fit into the memory of a single GPU. Finally, invariance to the ordering of the context allows CAMP to sidestep *recency bias* that skews in-context learning predictions towards the labels of recently seen demonstrations (Zhao et al., 2021).

CAMP similarly manifests advantages over classical gradient-based and metric-based meta-learning algorithms. Compared to gradient-based methods, CAMP does not require a fine-tuning step. Complicating factors such as overfitting to the support set, deciding which layers to reinitialize, or choosing fine-tuning hyperparameters become irrelevant. Similarly, the implementation complexities of metric-based approaches and the potential latency penalty incurred by computing distances—i.e. the Mahalanobis distance computes a per-class covariance matrix—are avoided.

## 4 THEORETICAL ANALYSIS

Invariance to permutations is an important property of few-shot learning algorithms. Informally, any permutation in the order of demonstrations should not affect the predictions of the model. This property is violated by in-context learning in LLMs, with recent work showing a bias towards demonstrations which occur in close proximity to the query (Zhao et al., 2021); however as it is invariant to permutations, CAMP is unaffected by this bias.

Kossen et al. (2021) show the Transformer encoder $f_\theta$ is permutation equivariant. For convenience, we repeat the properties, lemmas, and definitions developed in this work with language adapted to few-shot and in-context learning in Appendix A.1. It remains to show the output prediction of CAMP is invariant to permutations of the input sequence.

**Definition 1.** *Define the extraction function as $\mathbb{1}_i : \{S_1, ..., S_n\} \to S_i$ that extracts the $i^{th}$ element from a sequence.*

**Lemma 1.** *$\mathbb{1}_{\pi(i)}$ is permutation-invariant.*

*Proof.* For any permutation $\pi : [1, \ldots, n] \to [1, \ldots, n]$ applied to a sequence $\{x_i\}_{i=1}^n$,

$$\pi(\{x_i\}_{i=1}^n) = \{x_{\pi(i)}\}_{i=1}^n \tag{1}$$

and applying $\mathbb{1}_{\pi(i)}$ returns the original $i^{th}$ element:

$$\mathbb{1}_{\pi(i)} \circ \pi(\{x_i\}_{i=1}^n) = \mathbb{1}_{\pi(i)}(\{x_{\pi(i)}\}_{i=1}^n) = x_{\pi(i)} = \mathbb{1}_i(\{x_i\}_{i=1}^n) \tag{2}$$

$\square$

**Lemma 2.** *Let $f$ be a permutation-invariant function and $g$ be a permutation-equivariant function. Then $f \circ g(x)$ is permutation-invariant.*

*Proof.* Suppose not. Then $f(g \circ \pi(x)) \neq f(g(x))$ for some permutation $\pi$. But $g$ is permutation-equivariant, so $g(\pi(x)) = \pi(g(x))$. Similarly, $f$ is permutation-invariant so $f(g(\pi(x))) = f(\pi(g(x))) = f(g(x))$, hence contradiction. $\square$

**Property 4.0.1.** *The output prediction of CAMP is permutation-invariant.*

*Proof.* The output prediction of CAMP is $\text{MLP}(\mathbb{1}_i(f_\theta(\mathcal{S}^n)))$ where MLP stands for Multi-Layer Perceptron and $\mathcal{S}^n$ is the sequence of elements input to $f_\theta$. As the MLP takes as input a single vector, it is trivially permutation-invariant to the ordering of the input. Similarly, by Lemma 2 $\mathbb{1}_i(f_\theta(\mathcal{S}^n))$ is permutation-invariant as it is the composition of a permutation-invariation and a permutation-equivariant function. Therefore, $\text{MLP}(\mathbb{1}_i(f_\theta(\mathcal{S}^n)))$ is permutation-invariant with respect to the ordering of elements in the sequence. $\square$

## 5 ANALYSIS OF LEARNING MECHANISMS

In this section, we explore the learning mechanisms within CAMP and investigate how it uses the label information within a context of demonstrations to classify a query molecule.

**Dynamic Representations.** CAMP is conceptually similar to metric-based meta-learning in that both directly compare points in the support set with points in the query set. However unlike metric-based meta-learning, representations of points in the support set are not static. Rather, their learned representations are dynamically transformed alongside the query as this sequence is passed through the layers of a Transformer encoder. This is a very powerful feature; not only can the representations of the query dynamically adapt to relevant features within the demonstrations, but each demonstration can similarly dynamically adapt to relevant features within the query and other demonstrations from the context.

To visualize this dynamic, we plot joint label-molecule embeddings of 8 molecules from the BACE Classification dataset of the MoleculeNet benchmark (Wu et al., 2018) after dimensionality reduction by PCA (Pearson, 1901) in Figure 2a (left). Positively labeled points in the support set are represented by blue squares, negatively labeled points as red circles, and the query as a green triangle. The Transformer encoder of CAMP dynamically adapts support set and query representations in concert to build new representations of each. Figure 2a (center) depicts the Transformer encoder's output representations and highlights our findings that both the query and context output representations are linearly separable, even when linearly projected into a 2D space spanned by the principal components.

**Separation by Class Identity.** While it is unsurprising the model can separate demonstration vectors—after all, class identity is concatenated to the molecule embedding in the joint label-molecule space—it is perhaps surprising that the model actually does so. There is no explicit benefit from separating

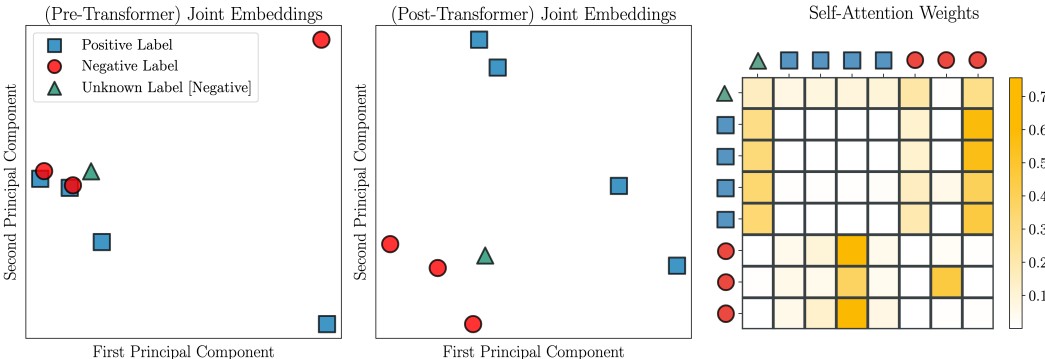

(a) Left: query and context joint embeddings before the Transformer encoder. Center: query and context joint embeddings after the Transformer encoder. Right: self-attention weights for a single head in the first transformer block. This matrix is row-orientated; $\mathcal{M}[i][j]$ depicts the weight with which the $i^{th}$ element of the sequence attends to the $j^{th}$ element.

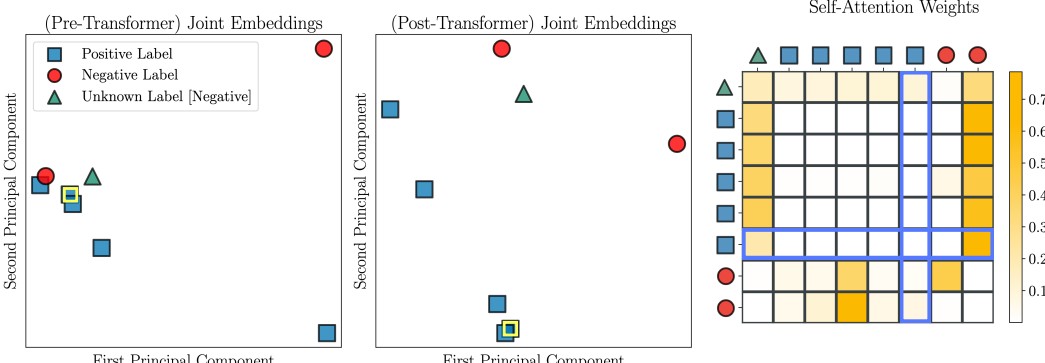

(b) Same as Figure 2a, but the $6^{th}$ element of the sequence—highlighted with a yellow outline—has its label flipped from negative to positive. The $6^{th}$ row and column of the self-attention weights matrix is outlined in blue to highlight how the self-attention weights change with relation to this point.

Figure 2: Visualization of learned representations for 8 molecules from the BACE dataset. PCA is used to reduce the hidden dimension of 768 to 2. The query vector is visualized with a green triangle, positive class demonstrations with blue squares, and negative class demonstrations with red circles. The true label of the query is negative.

demonstration vectors. No loss function or penalty is applied to demonstration vectors and they are discarded after the transformed query is extracted from the output of the Transformer encoder. Moreover, the query vector can be uniquely identified in the sequence by the [unknown] class embedding in place of a positive or negative label embedding.

To explore this characteristic, we examine the per-head attention weights learned by the model. Dissimilar to other self-attention weight visualizations, we observe a distinctive *vertical striation* pattern in many of the earlier layer self-attention mechanisms. In effect, many of the self-attention heads develop attention patterns specific to label identity, updating all demonstration vectors belonging to the same class in a similar manner. One such self-attention head is visualized in Figure 2a (right). In this example, all demonstration vectors with positive label strongly attend to the query and two negatively labeled demonstration vectors. Similarly, all demonstration vectors with a negative label strongly attend to the third positively labeled demonstration vector.

**Label Flip Analysis.** To further analyze these observations, we examine what would happen if we "flip" the label of a context vector. Figure 2b depicts our findings when we flip the label of the $5^{th}$ demonstration—highlighted in yellow—from negative to positive. As visualized in Figure 2b (left), the pre-encoder joint embeddings remain largely unchanged, simply changing a red circle to a blue square. However, the post-encoder embeddings are significantly different. Figure 2b (center) shows the $5^{th}$ demonstration vector switches sides from the linearly separated negatively labeled context vectors to the positively labeled ones. Interpreting this result, CAMP uses the label embedding

to separate demonstration vectors, irrespective to the molecule embedding dimensions in the joint label-molecule embedding space.

We can also examine how attention maps change. As visualized in the $6^{th}$ column of Figure 2b (right) and highlighted with blue outline, not only does the striation pattern persist, but it is also adapted to the new label distribution of the support set. As the label for the $5^{th}$ context vector flips from negative to positive, all positively labeled demonstrations no longer attend to the $5^{th}$ demonstration vector. Similarly, the attention pattern of the query vector changes. Previously, it attended to the $5^{th}$ and $7^{th}$ demonstrations, both negatively labeled, with approximately equal weight. After flipping the label, it attends to the $5^{th}$ demonstration with weight similar to that of other positively labeled demonstrations.

**Inferences.** To summarize, CAMP learns to linearly separate both query and context vectors, even when linearly projected into 2D space by PCA. Certain self-attention heads learn a *vertically striated* attention pattern based on label identity so that demonstrations with the same label have similar attention weights. Moreover, the query vector's attention pattern is similarly influenced by the class identity of demonstrations; although this change is less easily visualized.

Taken together, we posit CAMP leverages label identity to separate points in the support set and applies an analogous transformation to the query vector, developing representations keyed to the label identity of linearly separated demonstration vectors to determine the query's label. This hypothesis would explain why support set vectors become separated, even when there is no explicit gradient-based penalty to do so. It also accounts for the unique "vertical striation" pattern of many self-attention heads as transforming demonstration vectors with the same label to have similar representations may more easily pull the query vector to one of the two linearly separated groups. Further exploring this hypothesis, and perhaps replacing it with a more nuanced explanation, is an exciting direction for future work.

# 6 EXPERIMENTS

To analyze the effectiveness of CAMP, we evaluate its performance on the large-scale FS-Mol benchmark (Stanley et al., 2021) as well as the BACE Classification dataset from MoleculeNet (Wu et al., 2018) in the cross-domain setting.

## 6.1 FS-MOL EVALUATION

**Dataset Statistics.** FS-Mol is a large-scale, few-shot molecular property prediction benchmark composed of 5,120 binary classification tasks. Each task represents a protein, and the objective of the benchmark is to predict if a small molecule will activate (or inhibit) this protein. The dataset spans 233,786 different small molecules and the testing benchmark is composed of 157 protein targets.

**Experimental Design.** FS-Mol adheres to a typical few-shot learning paradigm. Models engage in large-scale pre-training on the train set, a held-out validation set is used for early stopping with a window-size of 10, and the model checkpoint with lowest validation loss is then evaluated on the test benchmark. We select Multi-Task Learning (MT), Model-Agnostic Meta-Learning (MAML) (Finn et al., 2017), Prototypical Networks (ProtoNet) (Snell et al., 2017), Conditional Neural Processes (CNP) (Garnelo et al., 2018), and Adaptive Deep Kernel Fitting with Implicit Function Theorem (ADKF-IFT) (Chen et al., 2022b) as our baselines. Another recent meta-learning baseline is MHNfs (Schimunek et al., 2023); however as this approach is complex: augmenting each support set with 25,000 context molecules and requiring 15,000 A100 GPU hours for hyperparameter tuning, we were unable to adapt their method to our experimental design. We structure each baseline to use Graph Neural Network (GNN) features by taking as input a connected graph of atom-level features.

While recent work has evaluated FS-Mol in a multi-modal setting by including ECFP features and/or PhysChem descriptors, we focus our empirical analysis on GNN features as they are the predominant modality used in real-world applications of few-shot molecular property prediction (Stokes et al., 2020; Liu et al., 2023; Wong et al., 2023). Additionally, computing ECFP fingerprints on-the-fly with RDKit slows down inference by approximately $1.5\times$. This latency penalty adds up when running

Table 1: Performance on 157 datasets of the FS-Mol few-shot learning benchmark. As in (Stanley et al., 2021), we report mean $\Delta$AUPRC as well as the standard error across 10 training runs.

| Approach | Aggregate $\Delta$AUPRC on FS-Mol | | | | |
|---|---|---|---|---|---|
| | 8 support | 16 support | 32 support | 64 support | 128 support |
| **Uses GNN Features** | | | | | |
| GNN-ST (Stanley et al., 2021) | — | $0.021 \pm 0.004$ | $0.027 \pm 0.005$ | $0.031 \pm 0.005$ | $0.052 \pm 0.005$ |
| MT Stanley et al. (2021) | $0.088 \pm 0.005$ | $0.112 \pm 0.006$ | $0.143 \pm 0.006$ | $0.176 \pm 0.008$ | $0.222 \pm 0.008$ |
| MAML Stanley et al. (2021) | $0.153 \pm 0.008$ | $0.160 \pm 0.009$ | $0.166 \pm 0.008$ | $0.173 \pm 0.008$ | $0.192 \pm 0.009$ |
| MAT Maziarka et al. (2020) | — | $0.051 \pm 0.005$ | $0.069 \pm 0.005$ | $0.092 \pm 0.007$ | $0.136 \pm 0.009$ |
| CNP Garnelo et al. (2018) | $0.174 \pm 0.010$ | $0.180 \pm 0.010$ | $0.186 \pm 0.010$ | $0.189 \pm 0.010$ | $0.205 \pm 0.010$ |
| ProtoNet Stanley et al. (2021) | $0.146 \pm 0.007$ | $0.185 \pm 0.008$ | $0.224 \pm 0.009$ | $\mathbf{0.256 \pm 0.009}$ | $\mathbf{0.290 \pm 0.009}$ |
| ADKF-IFT Chen et al. (2022b) | $0.079 \pm 0.005$ | $0.093 \pm 0.006$ | $0.108 \pm 0.007$ | $0.130 \pm 0.008$ | $0.177 \pm 0.009$ |
| CAMP | $\mathbf{0.205 \pm 0.009}$ | $\mathbf{0.229 \pm 0.009}$ | $\mathbf{0.246 \pm 0.010}$ | $\mathbf{0.257 \pm 0.010}$ | $0.273 \pm 0.010$ |

high-throughput screening on a dataset like the Enamine 9 billion REAL database, potentially turning weeks of processing into months.

With regards to evaluation criteria, we report $\Delta$AUPRC. $\Delta$AUPRC is the evaluation criteria employed by FS-Mol, and it measures the change of the area under the precision-recall curve with respect to a random classifier. This criteria is especially appropriate for highly unbalanced query sets, where the area under the precision-recall curve of a random classifier is equal to the percentage of positive examples in the query set. For unbalanced tasks, the relative change in performance is often more informative than the absolute area under the precision-recall curve.

**Training and Model Architecture** We follow the implementation and hyperparameters of Stanley et al. (2021) for the MT, MAML, and ProtoNet baselines and similarly adhere to the implementation and hyperparameters developed by Chen et al. (2022b) to benchmark CNP and ADKF-IFT. For the molecule encoder of CAMP, we adopt an MPNN Gilmer et al. (2017) architecture identical to the MT baseline, and the label encoder is a learnable linear projection from one-hot class labels. The Transformer encoder follows the base variant described by Dosovitskiy et al. (2020), and overall CAMP has 100.2 million parameters.

Given the size of the FS-Mol benchmark and limited computational resources, we mostly mimic the hyperparameters from the MT baseline when training CAMP but perform a small hyperparameter search over #warmup steps = $\{100, 2000\}$ and dropout = $\{0, 0.05, 0.1, 0.2\}$. Choosing #warmup steps = 2000 and dropout = 0.2 improves the validation loss from 0.588 with MT hyperparameters to 0.575. Further details related to reproducibility may be found in the Appendix, and our released code and model weights may be referenced to fully reproduce our results.

**Findings.** Table 1 summarizes our experimental findings across support sizes of $\{8, 16, 32, 64, 128\}$. To our surprise—and contrary to findings in the Natural Language Processing community comparing in-context learning with classic meta-learning algorithms (Liu et al., 2022; Perez et al., 2021; Min et al., 2021b)—CAMP outperforms all baselines at $\{8, 16, 32\}$ support sets. Moreover, it is competitive with the best performing approach, ProtoNet, at larger support sizes. Similarly surprising, the ADKF-IFT baseline significantly underperforms when evaluated with GNN features, even though this approach is state-of-the-art in the multi-modal setting.

## 6.2 BACE EVALUATION

**Dataset Statistics and Experimental Design.** BACE is a binary prediction task that measures if a small molecule will inhibit the human *beta-secretase 1* enzyme. The dataset contains measurements of 1,513 small molecules and could have been included as another task in the FS-Mol benchmark. We formulate each dataset as a cross-domain few-shot benchmark, randomly sampling a support set from each dataset and designating the remaining examples as the query set. All methods are meta-trained on FS-Mol similar to Section 6.1. We choose AUPRC, or the area under the precision-recall curve, to align our evaluation criteria with (Wu et al., 2018). Similar to our evaluation on FS-Mol, all models use GNN features, and we evaluate each support size across 10 different random seeds.

**Findings.** Our findings are presented in Table 2 and align with our empirical results on FS-Mol. While CAMP surpasses the performance of all baselines at each support size, this different tends to decrease as the cardinality of the support set increases. Specifically, CAMP surpasses the performance

Table 2: Performance on the BACE Classification dataset from the Molecule benchmark. We report mean AUPRC as well as the standard error across 10 training runs with different random seeds.

| Approach | Cross-Domain Meta-Learning AUPRC on BACE Classification | | | | |
|---|---|---|---|---|---|
| | 8 support | 16 support | 32 support | 64 support | 128 support |
| **Uses GNN Features** | | | | | |
| MT Stanley et al. (2021) | $0.581 \pm 0.038$ | $0.547 \pm 0.064$ | $0.600 \pm 0.063$ | $0.0652 \pm 0.051$ | $0.696 \pm 0.037$ |
| MAML Stanley et al. (2021) | $0.505 \pm 0.021$ | $0.51 \pm 0.026$ | $0.508 \pm 0.035$ | $0.548 \pm 0.064$ | $0.568 \pm 0.075$ |
| CNP Garnelo et al. (2018) | $0.508 \pm 0.026$ | $0.503 \pm 0.029$ | $0.496 \pm 0.021$ | $0.504 \pm 0.018$ | $0.507 \pm 0.009$ |
| ProtoNet Stanley et al. (2021) | $0.564 \pm 0.062$ | $0.594 \pm 0.057$ | $0.650 \pm 0.045$ | $0.690 \pm 0.021$ | $0.711 \pm 0.023$ |
| ADKF-IFT Chen et al. (2022b) | $0.528 \pm 0.045$ | $0.548 \pm 0.040$ | $0.570 \pm 0.058$ | $0.625 \pm 0.052$ | $0.677 \pm 0.013$ |
| CAMP | $\mathbf{0.651 \pm 0.033}$ | $\mathbf{0.686 \pm 0.033}$ | $\mathbf{0.700 \pm 0.030}$ | $\mathbf{0.728 \pm 0.011}$ | $\mathbf{0.737 \pm 0.009}$ |

Table 3: Inference-time latency as measured by wall-clock time on an A100 GPU to evaluate the 1,513 molecules in BACE across 16, 32, 64, and 128 support set sizes. Each evaluation is repeated 10 times across different random seeds. We exclude the 8-support as these experiments were run on a different server.

| | | MT | MAML | ProtoNet | CAMP | CNP | ADKF-IFT |
|---|---|---|---|---|---|---|---|
| Latency | Minutes & Seconds | 3m 11s | 3m 14s | 16m 47s | 3m 44s | 13m 17s | 14m 18s |
| | Normalized to CAMP | $0.85\times$ | $0.87\times$ | $4.50\times$ | $1\times$ | $3.56\times$ | $3.83\times$ |

of the next strongest baseline by $15\%$ at a support size of 8, by $15\%$ at 16, by $8\%$ at 32, by $6\%$ at 64, and by $9\%$ at 128.

## 6.3 INFERENCE-TIME LATENCY

Inference-time latency is critical for few-shot molecular property prediction as the most common application for these methods is high-throughput screening. With the space of pharmacologiaclly active compounds estimated to be on the order of $10^{60}$, and databases of order $10^{10}$ available for screening, even small differences in inference-time latency may lead to months of difference in screening times.

Each class of meta-learning algorithms has different inference-time behavior. Gradient-based methods require the model to be finetuned on the support set before query set examples can be evaluated, and metric-based methods require the computation of class centroids. Context-based methods sidestep both complexities, but instead require the full support set to be fed into the model alongside each query example. To quantify the extent to which these constraints affects inference-time latency, we offer a basic wall-clock time analysis of the time it takes to complete inference on the BACE dataset in Table 3. We note that our implementations are not optimized for serving, but our findings may offer a rough approximation to serving-time latency.

Our findings suggest MT, MAML, and CAMP all share roughly the same inference time latency. While the forward pass of CAMP is slower than that of MT and MAML, this difference is offset by the requirement that MT and MAML finetune the model's parameters on the support set. On the other hand, ProtoNet, ADKF-IFT, and CNP are significantly slower. Maximizing the GP marginal log likelihood with an L-BFGS (2nd order) optimizer during inference slows ADKF-IFT, and similarly, maximizing the conditional likelihood of a random subset of query molecules at each step of inference slows CNP. For ProtoNet, we attribute the increased latency to computing the full covariance matrix among support set points in the Mahalanobis distance calculation.

## 6.4 ABLATION STUDIES

**Naive ICL.** Our initial attempts at developing an in-context learning algorithm for few-shot molecular property prediction attempted to emulate the in-context learning paradigms found in LLMs. Notably, rather than concatenating the molecule embedding with the label embedding, we would interleave both along the sequence dimension and pass the entire sequence through a Transformer encoder. Despite many formulations of this common pattern, convergence would stall during the initial epochs of training, and although our best formulations would eventually begin converging, they would exhibit signs of instability throughout training. Figure 3 (left) displays the convergence of the best model from these studies, termed *Naive ICL*, and even though validation convergence is unstable, its

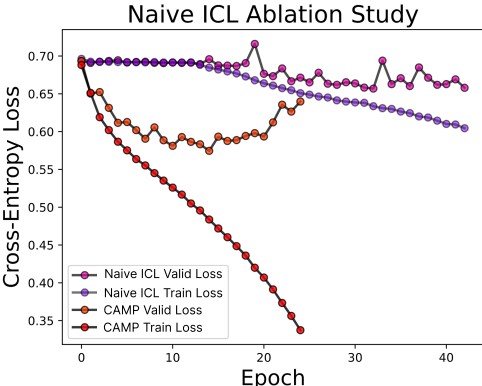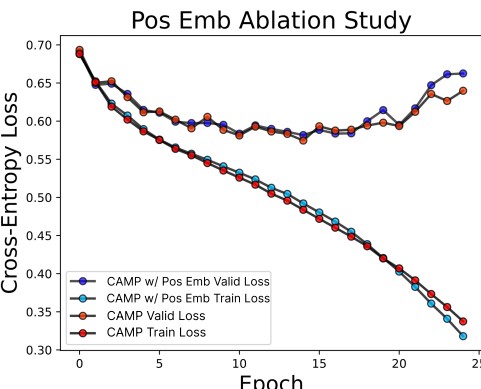

Figure 3: Left: convergence of Naive ICL compared to CAMP. Right: convergence of CAMP augmented with fixed positional embeddings (Vaswani et al., 2017) compared to no positional embeddings. For both plots, the full training run is displayed even though early stopping with a window size of 10 epochs is employed.

performance is comparable to that of gradient-based meta learning algorithms at small support splits when evaluated on FS-Mol.

**Positional Embeddings.** Our second study relates to the importance (or lack thereof) of positional embeddings. Static positional embeddings were found to be crucial to language modeling performance in Transformers (Vaswani et al., 2017) and learnable positional embeddings are similarly vital for Vision Trasformer (Dosovitskiy et al., 2020) and Audio-Spectrogram Transformer (Gong et al., 2021) performance. However, for the purpose of in-context learning, we question the relevancy of uniquely identifying the position of demonstrations in the context. The ordering of said examples is irrelevant, and while encoding knowledge related to the total number of examples into the learning process may be beneficial, rearranging the order of demonstrations in the context should not influence the model's predictions. Recent work has found the ordering of demonstrations to be problematic for in-context learning applications in natural language, biasing predictions towards the labels of demonstrations close to the query (Zhao et al., 2021).

We investigate this question in Figure 3 (right). CAMP without positional embeddings slightly outperforms the positional embeddings variant: $0.5746$ vs. $0.5817$ lowest validation loss. Moreover, adding positional embeddings may cause the model to overfit more quickly: $0.3374$ final training loss for CAMP vs. $0.3181$ for the variant with positional embeddings. This finding supports our hypothesis that the ordering of demonstrations within a context should be irrelevant to predictive performance.

## 7 CONCLUSION

In this work, we develop a new algorithm for few-shot molecular property prediction. Our approach reformulates the ideas underpinning in-context learning from a characteristic of LLMs to a few-shot learning algorithm that leverages a context of (molecule, property measurement) demonstrations to directly predict the property of a query molecule in a single forward pass. Our analysis suggests the learning mechanisms underpinning CAMP fundamentally differ from those in other Transformer-based architectures, and that it learns to classify query molecules by separating demonstration vectors by their class identities.

Further, our empirical evaluation indicates CAMP outperforms recent meta-learning algorithms at small support sizes. It is competitive with the best performing baseline, Prototypical Networks, at large support sizes while being approximately $5\times$ faster during inference. This result differs from comparisons in natural language that find the performance of typical meta-learning algorithms often surpasses that of in-context learning.

It is our hope that this work shows the benefits of in-context learning can be extended to applications outside of natural language. For instance, one might adapt our approach to few-shot image classification, replacing the molecule encoder with an image feature extractor. Other engaging avenues for future work may investigate its learning mechanisms or develop refinements to improve its performance at large support sizes.

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
