# OpenReview forum: "In-Context Learning for Few-Shot Molecular Property Prediction"
_ICLR.cc/2024/Conference — ICLR 2024 Conference Withdrawn Submission_

### Official Review · Reviewer_bDhz · 2023-10-31

**Soundness:** 3 good
**Presentation:** 2 fair
**Contribution:** 2 fair
**Rating:** 5
**Confidence:** 4

**Summary:**

This paper proposes an in-context learning framework for few-shot molecular property predictions. The CAMP framework embeds each (molecule, label) pair to encode the instance as context information for inferring labels of query molecules, which is transferring in-context learning techniques from LLM to FS-Mol tasks. Experimental results show the superiority of the proposed method.

**Strengths:**

(1) The basic idea of the proposed method is clearly illustrated and the fundamental structure of the paper is well-organized;

(2) The addressed few-shot molecular property prediction problem is an important challenge for drug discovery applications. This reviewer acknowledges the significance of the problem.

**Weaknesses:**

(1) The novelty of the proposed method is kind of limited. It seems the core idea is learning the relationships between support samples and query samples, which is very similar to PAR [1] in nature. PAR learns the relation graph across support samples and query samples. It seems this work only replaces the relation graph with the self-attention. The label encoding could be regarded as an alternative implementation of prototypes; Therefore, according to this reviewer's understanding, CAMP is just a small refinement over PAR;

(2) Although this work claims "in-context learning", it seems CAMP just focuses on the relation learning over support samples and query samples within a single task. If so, then the learning mechanism of CAMP is exactly the same as typical meta-learning frameworks. This reviewer thinks useful context information should be some additional information introduced from external sources. Also, "in-context learning" in LLM is used to demonstrate some examples to tell the large models what is the format of the target task, while the "task format" is not important in few-shot molecular property prediction.

(3) The experimental settings are not clearly introduced. First, this reviewer does not find which tasks in FS-Mol are selected to evaluate in this work. For example, ADKF-IFT and MHNF report their performance on some specific tasks including kinases, hydrolases, and oxidoreductases. Due to the large computational burden of MHNF, it is acceptable to only compare the proposed method with ADKF-IFT. However, the reported performance of ADKF-IFT is abnormally low compared to other baseline methods. This reviewer thinks the experimental tables in the paper are better aligned with the MHNF settings. Otherwise, the reported comparison is not that convincing;

[1] Property-aware relational networks for few-shot molecular property prediction. [NeurIPS 2021]

**Questions:**

(1) How does the proposed method tackle multi-task labels of molecules? Does each task label lead to an independent (molecule, label) pair? This part is still very ambiguous and this reviewer cannot figure out the exact mechanism of learning and inference;

(2) It seems the "in-context learning" only happens within a single sampled task, and then captures the relations between support samples and query samples. What are the differences between the proposed approach and PAR except for different encoder architectures?

---

### Official Review · Reviewer_69rR · 2023-11-01

**Soundness:** 2 fair
**Presentation:** 2 fair
**Contribution:** 1 poor
**Rating:** 3
**Confidence:** 4

**Summary:**

This paper proposes to handle few-shot molecular property prediction problem by in-context few-shot learning with transformer-based model. It follows a typical in-context learning way but the support molecules and labels and the queries are first mapped to vector by a MPNN and a linear embedding layer, then a standard transformer, and a MLP to predict. Empirical results show the proposed CAMP has advantage in performance with very small support size, and fast inference.

**Strengths:**

1.This paper is well-written.

2.This paper gives a solution for few-shot molecular property prediction with in-context learning. Empirical results on different datasets and analysis are provided.

**Weaknesses:**

The main concern is lack of novelty nor technical highlights. It looks like a simple combination of some existing approaches, directly applying the in-context learning in few-shot molecular property prediction problems. From the perspective of few-shot learning, such a standard  “encoder-transformer-mlp” framework can be applied for any few-shot classification problem, and ANALYSIS OF LEARNING MECHANISMS section proves it is very similar with the mechanism of [1]. From the perspective of molecular property prediction, no chemical prior knowledge is introduced, and the MPNN is a relatively old molecular encoder.

[1] Few-shot learning with graph neural networks, ICLR 2018.

**Questions:**

1. In-context learning is typically adopted with first large-scale pretraining. Could CAMP be improved by pretraining the encoder and transformer on auxiliary molecule datasets? The authors wrote that an in-context learning algorithm for few-shot molecular property prediction should "Train from random initialization on molecular property prediction datasets", why? There already exist large scale datasets such as ZINC15 and Chembl, which can be used to pretrain the model.

2. Can you describe the main difference with your learning mechanisim w.r.t PAR [2]? In PAR, it also satisfies that "learned representations are dynamically transformed alongside the query". Figure 2 in this paper also shares the same spirit of Figure 5 and Figure 6 in PAR.

3. In Table 3: Inference-time. Why MAML can inference quickly as it requires several gradient steps on support set? Why ProtoNet and CNP inference much slower than CAMP, as they all take only one forward-propagation to adapt to the task?

[2] Property-aware relation networks for few-shot molecular property prediction, NeurIPS 2021.

---

### Official Review · Reviewer_62tm · 2023-11-05

**Soundness:** 2 fair
**Presentation:** 2 fair
**Contribution:** 1 poor
**Rating:** 3
**Confidence:** 4

**Summary:**

This paper tackles the problem of few-shot molecular property prediction. It proposes an in-context learning for few-shot molecule property prediction. It concatenates the molecular and label embeddings and feeds the vectors into the Transformer encoder for the classification prediction. The framework makes it possible to deal with the new tasks without fine-tuning steps. The effectiveness of the proposed method has been demonstrated on two benchmarks.

**Strengths:**

1.	The studied problem is interesting and important. The clarity of writing and structure of the paper effectively convey the methodology to the reader.
2.	The proposed method is simple but effective, enabling in-context learning as a viable approach for molecular property prediction.

**Weaknesses:**

1.	The proposed framework seems to overlap significantly with another anonymous submission[1], especially Figure 1 in both papers. Not sure whether they are from the same authors, but it clearly dilutes the technical contribution of the current work.
2.	Missing critical baselines. Given that few-shot molecular property prediction is not a new research topic, there are some previous works proposed specifically for this problem, such as [2]. It is crucial to include the comparisons with these baselines to establish the relative performance and advancement of the proposed method.
3.	The experimental setting is not reasonable. While the introduction asserts that the CAMP framework can quickly adapt to new tasks without fine-tuning, this adaptability cannot be adequately demonstrated in the experimental design. The BACE dataset has only a single classification task. In FS-Mol datasets, despite containing multiple classification tasks, the absence of a task-specific split in the dataset precludes testing on truly new tasks.
4.	The experiments are not sufficient. There are only two benchmark datasets used in the experiments. Although BACE is a recognized benchmark in the MoleculeNet[3], there are many other benchmarks available. Some of these datasets have multiple classification tasks in one dataset, which is probably more suitable for evaluating the proposed method.
5.	Figure 3 is very difficult to read because of the chosen colors.

[1] Context-Aware Meta-learning. https://openreview.net/pdf?id=lJYAkDVnRU
[2] Property-aware relation networks for few-shot molecular property prediction, NeurIPS 2021.
[3] https://moleculenet.org/datasets-1

**Questions:**

See weaknesses.

---

### Official Review · Reviewer_8K7Q · 2023-11-08

**Soundness:** 2 fair
**Presentation:** 3 good
**Contribution:** 2 fair
**Rating:** 3
**Confidence:** 4

**Summary:**

This paper studies few-shot molecular property prediction with an in-context learning formulation. A method called Context Aware Molecule Prediction (CAMP) is proposed, which employs a transformer encoder to learn representations for the set of (molecule, property) pairs from both support and query sets and an MLP head to predict the property of the query molecule based on its transformed representation. The proposed method is evaluated on the FS-Mol and BACE binary classification benchmarks. Ablation studies are conducted to explore the effects of positional embedding and (molecule, property) embedding concatenation.

**Strengths:**

1. The formulation of in-context learning for molecular property prediction is interesting as the transformer encoder learns to extracts dynamic representations for the support (molecule, property) pairs alongside the query molecules, which is more powerful than the static representations in metric-based methods and more efficient than gradient-based methods.
2. The analysis and visualization of the embeddings and attention weights of CAMP are helpful for understanding the properties of the learned representations.

**Weaknesses:**

1. My main concern is the experiments. In the abstract, the paper claims that
> On the FS-Mol and BACE molecular property prediction benchmarks, we find this method surpasses the performance of recent meta-learning algorithms at small support sizes and is competitive with the best methods at large support sizes.

However, the above claim is not an accurate description of the results in the paper, because the experimental setup in Section 6.1 is different from the setup of the FS-Mol benchmark. Specifically, the results in Section 6.1 are obtained with GNN features while the original FS-Mol benchmark and all subsequent works consider GNN + ECFP + PhysChem descriptors. I suspect this is why the performance of ADKF-IFT reported in Section 6.1 is significantly worse than that reported in the original paper [1]. The authors justify the removal of the ECFP and PhysChem descriptors by the observation that computing these features significantly slows down the inference speed, which does not seem to make sense because these features should be cheap to compute (actually much faster than computing the GNN features). Also, as far as I know, ECFP in particular is still the No.1 choice in industry. Therefore, the comparison would be more meaningful and useful if all methods are run with the original setup from the FS-Mol paper [2].

2. While the in-context learning formulation for molecular property prediction is interesting, the technical contribution of this work is quite limited as it is just an application of the transformer encoder to the few-shot molecular property prediction problem. The idea of concatenating (molecule, property) embedding (i.e., concatenating (x, y) embedding) and the permutation invariance property of transformers have already been thoroughly investigated in the set transformer paper [3].

3. There are a few questions that need to be clarified. See the Questions section below.


[1] Wenlin Chen, et al. "Meta-learning adaptive deep kernel gaussian processes for molecular property prediction." ICLR 2023.

[2] Megan Stanley, et al. "Fs-mol: A few-shot learning dataset of molecules." NeurIPS 2021 (Datasets and Benchmarks Track).

[3] Juho Lee, et al. "Set transformer: A framework for attention-based permutation-invariant neural networks." ICML 2019.

**Questions:**

1. For the visualizations of the self-attention weights in Figure 2, the vertical striation pattern is interesting. Can the authors explain why there is such a vertical striation pattern in the self-attention weights? Intuitively, each molecule should attend to the ones that have the same property label as its own label, but it does not seem to be the case in the visualization.

2. It is sensible that ADKF-IFT is slower than CAMP due to the inner loop optimization of the GP parameters. However, can the authors explain why CNP and ProtoNet are much slower than CAMP as reported in Table 3? This does not seem to make sense because none of these methods require parameter updates at inference time. In theory, the inference time cost of CAMP is $O(N^2)$ due to the self-attention computation, which is the same as computing the Mahalanobis distance for each pair of molecules (also $O(N^2)$) in ProtoNet. CNP should be even faster at inference time as it only costs $O(N)$.